# Detecting Hands and Recognizing
# Physical Contact in the Wild

**Supreeth Narasimhaswamy**[1]    **Trung Nguyen**[2]    **Minh Hoai**[1,2]
[1]Stony Brook University, Stony Brook, NY 11790, USA
[2]VinAI Research, Hanoi, Vietnam
`{sunarasimhas, minhhoai}@cs.stonybrook.edu`

## Abstract

We investigate a new problem of detecting hands and recognizing their physical contact state in unconstrained conditions. This is a challenging inference task given the need to reason beyond the local appearance of hands. The lack of training annotations indicating which object or parts of an object the hand is in contact with further complicates the task. We propose a novel convolutional network based on Mask-RCNN that can jointly learn to localize hands and predict their physical contact to address this problem. The network uses outputs from another object detector to obtain locations of objects present in the scene. It uses these outputs and hand locations to recognize the hand's contact state using two attention mechanisms. The first attention mechanism is based on the hand and a region's affinity, enclosing the hand and the object, and densely pools features from this region to the hand region. The second attention module adaptively selects salient features from this plausible region of contact. To develop and evaluate our method's performance, we introduce a large-scale dataset called ContactHands, containing unconstrained images annotated with hand locations and contact states. The proposed network, including the parameters of attention modules, is end-to-end trainable. This network achieves approximately 7% relative improvement over a baseline network that was built on the vanilla Mask-RCNN architecture and trained for recognizing hand contact states. Code and data are available at: `https://github.com/cvlab-stonybrook/ContactHands`.

## 1  Introduction

The objective of this work is to detect hands in images and recognize their physical contact state. By physical contact state, we mean to recognize the following four conditions for each hand instance, namely (1) No-Contact: the hand is not in contact with any object in the scene; (2) Self-Contact: the hand is in contact with another body part of the same person; (3) Other-Person-Contact: the hand is in contact with another person; and (4) Object-Contact: the hand is holding or touching an object other than people. These conditions are not mutually exclusive, and a hand can be in multiple states; for example, a hand can contact another person and, at the same time, hold an object. Detecting hands and recognizing their physical contact is an important problem with many potential applications, including harassment detection, contamination prevention, and activity recognition.

However, recognizing the contact state of a hand in unconstrained conditions is challenging because the hand's appearance alone is insufficient to estimate its contact state. This task also requires us to consider the relationships between the hand and other objects in the scene. This can be a complex inference problem for many real-world situations, especially where numerous people and objects surround the hand. Furthermore, even for a pair of hand and object with corresponding segmentation masks, it is not easy to recognize whether the hand is in contact with the object due to the lack of

depth information. A heuristic-based method using occlusion or overlapping criteria would not work well because the hand can hover in front of the object without touching it.

In this work, we propose a Contact-Estimation neural network module for recognizing the physical contact state of hands. This module can be integrated into an object detection framework to detect hands and recognize their contact states jointly. Together with the hand detector, we can train the Contact-Estimation module end-to-end using training images where hands are localized and annotated with corresponding contact states. Notably, our method does not require annotation for the contact object or contact areas. One technical contribution of our paper is learning to recognize contact states using such weak annotations.

Specifically, we implement our method based on Mask-RCNN [13], a state-of-the-art object detection framework. Mask-RCNN has a Region Proposal Network (RPN) that first generates a candidate hand proposal box. A box regression head and a mask head then obtain the bounding box and a binary segmentation map of the hand. Additionally, we obtain the locations of other objects in the scene using a generic object detector pre-trained on the COCO [18] dataset. We then use the Contact-Estimation branch to recognize the contact state for detected hands. The inputs to this new branch are: (1) the feature maps for the hand, and (2) a set of $K$ feature maps, one for each hand-object union box, where $K$ is the number of detected objects.

Given the above inputs, we use the Contact-Estimation network module to compute scores for each of the $K$ hand-object pairs. We first combine the hand feature map with the hand-object union feature map at particular spatial locations. Intuitively, if the location $A$ of the hand is in contact with the location $B$ of the object, it would be useful to combine hand features at $A$ with the object features at $B$. We formalize this notion using a cross-feature affinity-based attentional pooling module that can combine hand and hand-object union features from various locations based on the affinities between them. Second, the hand-object union feature map encodes the regions between the hand and the object and can contain possible contact regions. We propose a spatial attention method to learn to focus on salient regions. Finally, we obtain contact state scores for each of the $K$ hand-object pairs independently using the cross-feature affinity-based attention module and spatial attention module. The proposed attention modules are trained end-to-end together with the Contact-Estimation branch.

Another contribution of our paper is a large-scale dataset for development and evaluation. Our dataset consists of around 21K images, containing bounding box annotations for 58K hands and their physical contact states. The dataset contains many challenging images in the wild, where it is not trivial to determine the physical contact states of hands. This dataset can be used to develop real-world applications that require contact states of hands, such as contamination prevention and harassment detection.

## 2   Related Work

We build upon two-stage object detection frameworks such as [10, 11, 13, 23]. The current object detection frameworks recognize an object's presence or absence at a particular region of interest by classifying the pooled feature inside this region. However, such a framework is insufficient for our problem. In our case, we need to detect hands and recognize their physical contact state by reasoning about other surrounding objects.

There are prior works for hand detection [3, 6, 7, 15–17, 19, 21, 22, 26, 28, 33, 35], hand pose estimation [12, 25, 29, 37, 38], and hand-tracking [3]. However, they do not recognize the physical contact state of hands. One way to estimate the contact state of a hand is to reason based on its estimated pose. However, obtaining a reasonably good hand pose in unconstrained conditions is itself a challenging task. For example, obtaining the hand pose in low-resolution visual data such as surveillance images in a super-market or an elevator is highly difficult. Therefore recognizing physical contact states of hands using their pose is not a robust approach.

Some works consider hand-object interactions. Hasson et al. [12] propose to reconstruct hand and object meshes from monocular images. Bambach et al. [2] aim to locate hands interacting with an object, focusing on first-person videos containing only two people. Moreover, most of the activities focus on hands playing cards or puzzles. Tekin et al. [30] propose to model hand-object interactions by jointly estimating the 3-D poses for hand and objects. However, they only consider first-person views. More importantly, these methods do not estimate the physical contact state of the hand.

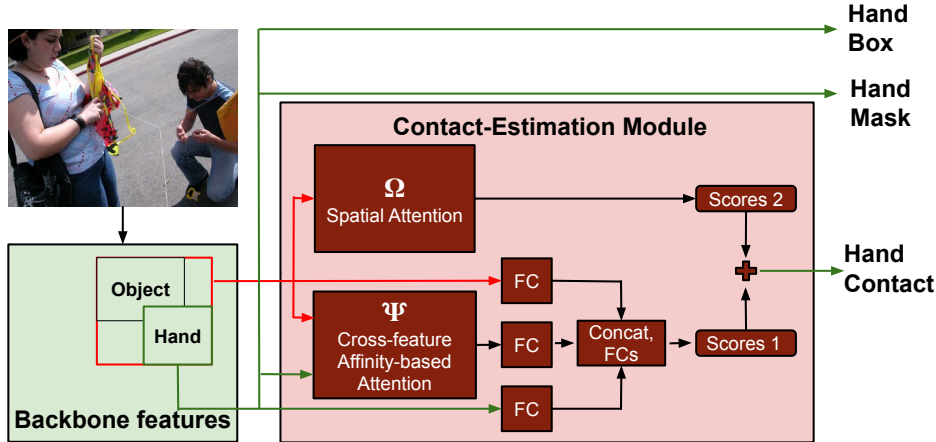

Figure 1: **Processing pipeline for joint hand detection and contact state recognition.** The bounding box regression head and mask head use the hand feature map to generate the hand's bounding box and mask. The Contact-Estimation module takes the hand feature map and hand-object union feature map as inputs. The cross-feature affinity-based attentional pooling pools hand-object union features to the hand features. The spatial attention method focuses on selective regions in the hand-object union feature map.

Closely related to our problem is the work from Shan et al. [27]. They propose a video-frame dataset of everyday interactions scraped from Youtube and annotate them with hand locations, hand side, contact state, and contact object location. In our work, we aim to recognize physical contact in the wild, and the images in our dataset are unconstrained. Shan et al. [27] also developed a method using Faster-RCNN to detect hands and predict contact based on the hand's appearance. Our method instead predicts hand's contact by considering hand and other surrounding objects. Another notable difference is that our method does not assume that a hand can only be in one contact state. [27] treats contact recognition as a multi-class classification problem. Instead, we treat it as a multi-label classification problem and train our method using four independent binary cross-entropy losses, one for each of the four possible contact states.

Our method consists of two attention mechanisms. The spatial attention method shares similarities with several visual attention methods that have gained much interest over the past years [1, 5, 9, 14, 20, 21, 24, 31, 32, 36]. The cross-feature affinity-based attentional pooling is inspired by [32]. We design it to pool hand-object union features to hand features by considering affinities between them at every spatial location.

## 3 Approach

In this section, we will describe our framework's overall architecture and provide details of the Contact-Estimation module used to recognize the physical contact state of a hand.

### 3.1 Model Overview

The proposed architecture is illustrated in Figure 1. A Region Proposal Network (RPN) obtains rectangular hand proposals. For each proposal, we extract ResNet backbone features of dimensions $h \times w \times d$ using the RoI Align operation and perform the bounding box regression and the binary mask segmentation. Additionally, we use a pre-trained object detector to detect other objects in the image. We use such detected objects to obtain hand-object union regions. Suppose the number of detected objects is $K$. We then extract $K$ ResNet features of dimensions $h \times w \times d$, one for each $K$ hand-object union regions. These features, together with the hand features, are then passed to the Contact-Estimation module. The Contact-Estimation module then computes 4-dimensional score vectors $\mathbf{s}_k \in \mathbb{R}^4, 1 \leq k \leq K$, one for each $K$ hand-object pairs. The $K$ score vectors are then combined to a single vector $\mathbf{s} \in \mathbb{R}^4$ to obtain the contact state class scores for the hand. We now provide more details about computing the contact state class scores $\mathbf{s} \in \mathbb{R}^4$ from the hand features $\mathbf{H} \in \mathbb{R}^{h \times w \times d}$ and $K$ hand-object union features $\mathbf{U}_1, \cdots, \mathbf{U}_K \in \mathbb{R}^{h \times w \times d}$.

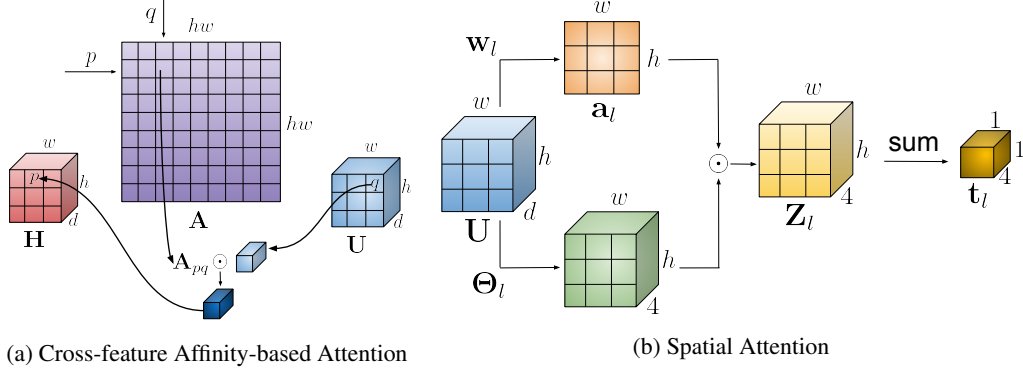

(a) Cross-feature Affinity-based Attention      (b) Spatial Attention

Figure 2: **(a) Cross-feature Affinity-based Attentional Pooling.** We pool the hand-object union feature from $\mathbf{U}$'s $q^{th}$ location to the hand feature $\mathbf{H}$'s $p^{th}$ location, weighted by the affinity $\mathbf{A}_{pq}$ between them. We do this densely for all spatial locations $p$ and $q$. **(b) Spatial Attention.** The attention map $\mathbf{a}_l$ encodes salient regions of the hand-object union region. We use $\mathbf{a}_l$ to select scores from such locations to obtain $\mathbf{Z}_l$. We finally obtain the scores $\mathbf{t}_l$ by summing scores from all spatial locations of $\mathbf{Z}_l$.

## 3.2 Recognizing Physical Contact using Multiple Objects

We now present the forward logic for the Contact-Estimation module. It takes two sets of inputs, the hand feature map $\mathbf{H} \in \mathbb{R}^{h \times w \times d}$ and $K$ hand-object union feature maps $\mathbf{U}_1, \cdots, \mathbf{U}_K \in \mathbb{R}^{h \times w \times d}$, one for each $K$ detected objects. The output of this module is a vector of contact state scores $\mathbf{s} \in \mathbb{R}^4$.

For each hand-object pair $k$, $1 \le k \le K$, we first obtain a vector of scores $\mathbf{s}_k \in \mathbb{R}^4$ as follows:

1. We obtain features $\Psi(\mathbf{H}, \mathbf{U}_k) \in \mathbb{R}^{h \times w \times d}$ by combining hand-object union features $\mathbf{U}_k$ with the hand features $\mathbf{H}$ using the cross-feature affinity-based attentional pooling module $\Psi$.

2. We pass features $\Psi(\mathbf{H}, \mathbf{U}_k), \mathbf{H}, \mathbf{U}_k$ through fully-connected layers, concatenate them, and finally project using fully-connected layers to obtain a first set of scores $\mathbf{s}_k^{(1)} \in \mathbb{R}^4$.

3. We obtain a second set of scores $\mathbf{s}_k^{(2)} := \Omega(\mathbf{U}_k) \in \mathbb{R}^4$ by passing the hand-object union feature map $\mathbf{U}_k$ through the spatial attention module $\Omega$.

4. We compute the class scores $\mathbf{s}_k \in \mathbb{R}^4$ for the $k^{th}$ hand-object pair as $\mathbf{s}_k := \mathbf{s}_k^{(1)} + \mathbf{s}_k^{(2)}$.

Once we obtain scores $\mathbf{s}_k \in \mathbb{R}^4$ for each $K$ hand-object pairs, we compute the contact state scores $\mathbf{s} \in \mathbb{R}^4$ for the hand feature $\mathbf{H}$ by taking the element wise maximum of $K$ scores: $\mathbf{s} := \max_{1 \le k \le K} \mathbf{s}_k$.

## 3.3 Cross-feature Affinity-based Attentional Pooling to Combine Features

We now describe a method to combine hand features with object features. Based on the intuition that different regions of the object have different affinities to contact the hand, we propose an attention method that combines features at each spatial location of the hand with features from all possible spatial locations of the hand-object union region, weighted by affinities. We parameterize these affinities in terms of the attention module's weights and learn them end-to-end during training. Fig. 2a illustrates this attention method.

The attention module takes as input the hand features $\mathbf{H} \in \mathbb{R}^{n \times d}$ and the hand-object union features $\mathbf{U} \in \mathbb{R}^{n \times d}$. Here, $n := hw$ denotes the number of spatial locations and $d$ denotes each feature's dimensions. The attention module outputs combined features $\Psi(\mathbf{H}, \mathbf{U}) \in \mathbb{R}^{n \times d}$ as:

$$\Psi(\mathbf{H}, \mathbf{U}) := \mathbf{H} + \text{softmax}(\mathbf{A})\mathbf{U}. \tag{1}$$

Here, $\mathbf{A} \in \mathbb{R}^{n \times n}$ is a matrix such that $\mathbf{A}_{p,q}$ encodes the affinity between the $p^{th}$ hand features and the $q^{th}$ hand-object union features and softmax($\mathbf{A}$) denotes the softmax taken along the last dimension of $\mathbf{A}$. We parameterize $\mathbf{A}$ using weights $\mathbf{W}_\alpha \in \mathbb{R}^{d \times d}$ and $\mathbf{W}_\beta \in \mathbb{R}^{d \times d}$ as follows:

$$\mathbf{A} := (\mathbf{H}\mathbf{W}_\alpha)(\mathbf{U}\mathbf{W}_\beta)^T. \tag{2}$$

We implement the weights $\mathbf{W}_\alpha$ and $\mathbf{W}_\beta$ using $1\times1$ convolutions and learn them during training. Notably, we can implement this attention module's entire forward logic as a few matrix multiplications and additions in less than five lines of PyTorch code.

## 3.4 Spatial Attention to Learn Salient Regions

The hand-object union feature map encodes both the hand and the object's appearance and can contain crucial contextual regions that determine the hand's physical contact state. However, it is not trivial to select features from such regions. This subsection proposes an attention method to learn salient regions adaptively and predict contact state scores based on such regions' features. The attention module takes as input the hand-object union features $\mathbf{U} \in \mathbb{R}^{n \times d}$, where $n := hw$ denotes the number of spatial locations and $d$ denotes feature's dimensions. The output of the attention module is a vector of contact state class scores $\mathbf{s}^{(2)} = \Omega(\mathbf{U}) \in \mathbb{R}^4$.

To recognize the physical contact state of the hand, we first localize the areas of the hand-object union region that are relevant for the recognition decision. Specifically, we learn $L$ spatial attention maps $\mathbf{a}_1, \cdots, \mathbf{a}_L \in \mathbb{R}^n$ that focus on selective regions of the hand-object union features. Corresponding to each such attention map $\mathbf{a}_l$, $1 \leq l \leq L$, we obtain score vector $\mathbf{t}_l \in \mathbb{R}^4$. We do this by predicting score vectors at each spatial location of the hand-object union region and averaging them, weighted by the attention map. We finally obtain the contact state scores $\mathbf{s}^{(2)}$ by averaging score vectors $\mathbf{t}_1 \cdots, \mathbf{t}_L$ corresponding to all $L$ attention maps. We illustrate the proposed attention method in Fig. 2b.

Formally, we first define the attention maps $\mathbf{a}_1, \cdots, \mathbf{a}_L \in \mathbb{R}^n$ by $\mathbf{a}_l := \mathrm{softmax}(\mathbf{U}\mathbf{w}_l)$.     (3)

Here, $\mathbf{w}_l \in \mathbb{R}^d$ is a learnable weight vector for the $l^{th}$ attention map $\mathbf{a}_l$.

Next, for each attention map $\mathbf{a}_l$, we define $\mathbf{Z}_l \in \mathbb{R}^{n \times 4}$ as $\mathbf{Z}_l := \mathbf{a}_l \odot (\mathbf{U}\mathbf{\Theta}_l)$.     (4)

Here, $\mathbf{\Theta}_l \in \mathbb{R}^{d \times 4}$ are learnable weights and $\odot$ denotes the element-wise multiplication by broadcasting elements of $\mathbf{a}_l$. Intuitively, $\mathbf{Z}_l$ encodes scores at all $n$ spatial locations weighted by the $l^{th}$ attention map $\mathbf{a}_l$. We then compute the score vector $\mathbf{t}_l \in \mathbb{R}^4$ corresponding to the $l^{th}$ attention map $\mathbf{a}_l$ by summing scores at all spatial locations of $\mathbf{Z}_l$.

Finally, we compute the contact state class scores $\mathbf{s}^{(2)} \in \mathbb{R}^4$ by averaging scores $\mathbf{t}_1, \cdots, \mathbf{t}_L$ corresponding to all $L$ attention maps $\mathbf{a}_1, \cdots, \mathbf{a}_L$: $\mathbf{s}^{(2)} := (\sum_{l=1}^{L} \mathbf{t}_l)/L$.

We implement the weights $\mathbf{w}_l$ in Eq. (3) and $\mathbf{\Theta}_l$ in Eq. (4) as $1\times1$ convolutions and learn them end-to-end during training. The entire arithmetic operations involved can be implemented as vectorized operations within ten lines of PyTorch code.

## 3.5 Loss Function for the Proposed Architecture

We train the entire network containing the bounding box regression, mask generation, and contact-estimation branches end-to-end by jointly optimizing the following multi-task loss:

$$\mathcal{L} := \mathcal{L}_{cls} + \mathcal{L}_{box} + \mathcal{L}_{mask} + \lambda\mathcal{L}_{contact}. \qquad (5)$$

Here, $\mathcal{L}_{cls}$, $\mathcal{L}_{box}$, $\mathcal{L}_{mask}$ denote the classification, the bounding box regression, and the segmentation mask losses, respectively. These are the standard loss terms of the Mask-RCNN object detection framework [13]. The term $\mathcal{L}_{contact}$ denotes the loss for the physical contact state of the hand, and $\lambda$ is a tunable hyperparameter denoting the weight of the contact loss. The contact loss $\mathcal{L}_{contact}$ is the sum of four independent binary cross-entropy losses corresponding to four possible contact conditions, i.e., $\mathcal{L}_{contact} := L_1 + L_2 + L_3 + L_4$. We define the contact loss based on independent binary cross-entropy losses instead of a single softmax cross-entropy loss since a hand can have multiple contact conditions. Thus, it is better to treat contact recognition as a multi-label classification problem rather than a multi-class classification.

## 4 ContactHands Dataset

We collect a large-scale dataset of unconstrained images for the development and evaluation of our model. We aim to collect images containing diverse hands with various shapes, sizes, orientations,

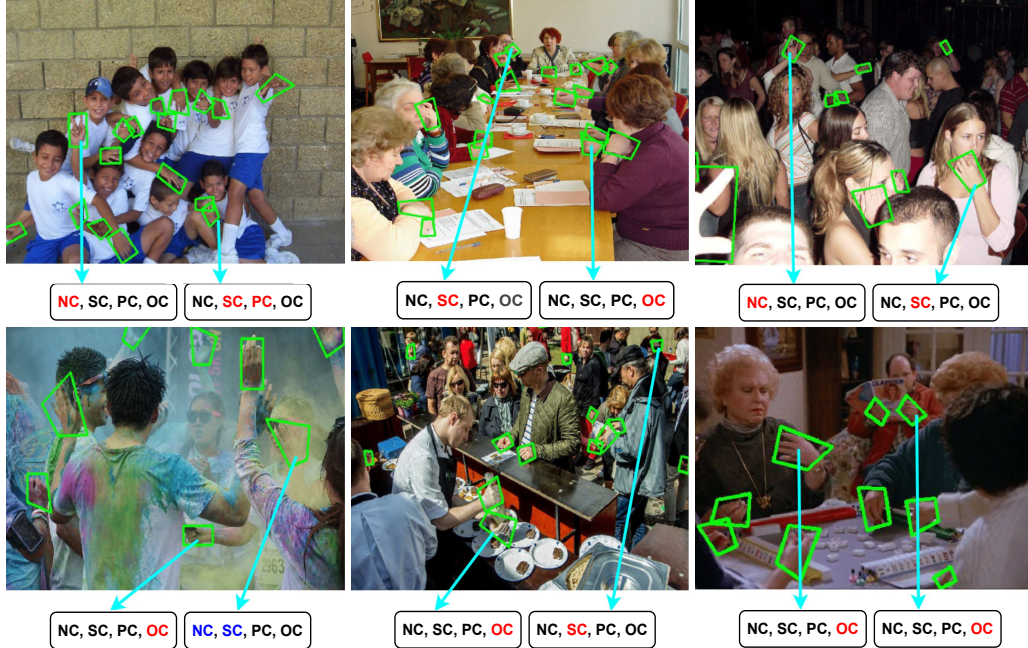

Figure 3: **Sample data from ContactHands.** We show the bounding box annotations in green color. To avoid clutter, we display contact states for only two hand instances per image. The notations NC, SC, PC, and OC denote No-Contact, Self-Contact, Other-Person-Contact, and Object-Contact. We highlight the contact state for a hand by red color. If a contact state is unsure, we highlight it in blue.

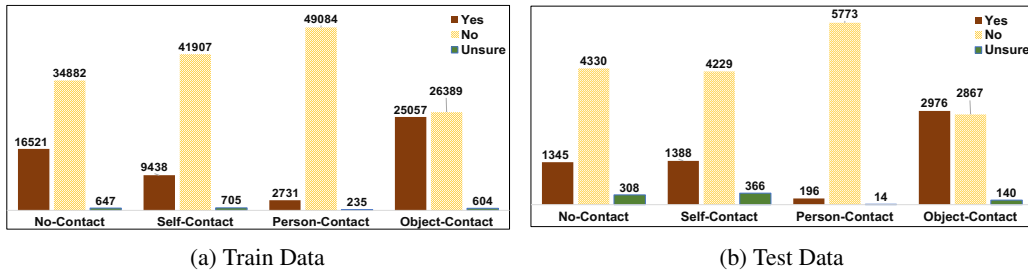

(a) Train Data           (b) Test Data

Figure 4: **ContactHands dataset statistics.** There are 52,050 and 5,893 annotated hand instances in the training and the test set. For each hand instance, we provide contact state annotations by choosing *Yes*, *No*, or *Unsure*.

and skin tone. We name the proposed dataset ContactHands, and it has numerous hand instances for which it is challenging to recognize the contact state.

**Dataset Source.** We collect two types of data, still photos and video frames. For still images, we collect images from multiple sources. First, we select images that contain people from popular datasets such as MS COCO [18] and PASCAL VOC [8] datasets. The COCO dataset images have everyday objects containing various annotations, including bounding boxes, keypoints, and segmentation masks for persons. Similarly, the PASCAL VOC is a benchmark in the visual object category recognition and object detection. However, these datasets do not have annotation for hand bounding boxes and their physical contact states. As a second source for still photos, we scrape some photos from Flickr using keywords that are likely to return pictures containing people. For example, we use keywords such as *people*, *cafeteria*, *parks*, *party*, *shopping*, *library*, *students*, *camping*, *vacation*, *outdoors*, *meeting*, *hanging-out*, *tourists*, and *festivals*. We only collect those pictures which have appropriate copy-right permissions. Also, we manually inspect each image to keep only those that contain at least one person. Together with the MS COCO and PASCAL VOC dataset images, these images form our still images group. To complement the still photos, we also collect frames sampled from videos. For this purpose, we use the training and validation split of the Oxford Hand dataset [19] and TV-Hand [21] dataset. Altogether, our dataset has 21,637 images.

**Annotation and Quality Control, Dataset Statistics.** We annotate the data using multiple annotators and subsequently verify them. We ask annotators to localize hand instances by drawing a tight quadrilateral bounding box that contains as many hand pixels as possible. We instruct them to localize all hands for which the minimum side of the resulting axis-parallel box has a length greater than $\min(H, W)/30$, where $H$ and $W$ are the height and width of the image. We ask annotators to localize truncated and occluded hands as long as the visible hand areas' size is greater than the previously mentioned threshold. We choose quadrilateral boxes instead of axis-parallel bounding boxes since hands are incredibly articulate, and the axis-parallel bounding box provides poor localization for hands. In addition to localizing hands, we ask annotators to identify the physical contact state for each hand instance. Since hands can be in multiple contact states, we instruct the annotators to consider the four possible contact states independently; we ask them to answer *Yes*, *No*, and *Unsure* for each of the four possible contact states separately.

We collect annotations in batches. We ask an additional annotator to verify annotations for quadrilateral boxes and contact states for each batch. We further verify each batch's physical contact states' annotations by randomly sampling a fraction of images and independently annotating contact states for every hand instance. We then quantitatively measure the error in annotations to verify that the error is within 2% for all annotations batches. Figure 3 shows some sample images and annotations from our dataset ContactHands.

The total number of annotated hand instances are 58,165. We randomly sample 18,877 images from these annotated images to be our training set and 1,629 images to be our test set. There are 52,050 and 5,983 hand instances in train and test sets, respectively. Figure 4 displays some statistics about contact states annotations.

## 5  Experiments

In this section, we will provide details about model implementation and hyperparameters. We will also explain the evaluation metric and present experimental results.

**Model Implementation and Hyperparameters.** We implement the proposed architecture using Detectron2 [34]. We add a Group Normalization layer before the residual connection in the cross-feature affinity-based attentional pooling to stabilize the training. We set the number of attention maps $L$ for the spatial attention module to be 32. The weight $\lambda$ for the contact state loss $\mathcal{L}_{contact}$ in the Eq.( 5) is set to 1. The binary cross-entropy losses for all four contact states have equal weights; i.e., we do not scale the losses. The fully-connected layers in the Contact-Estimation branch have dimensions 1024. Note that tuning the loss weights for four states, parameter $L$, and dimensions for fully-connected layers can likely give better results. We train the network using SGD with an initial learning rate of 0.001 and a batch size of 1. We reduced the learning rate by a factor of 10 when the performance plateaued. We do not penalize contact state predictions for *Unsure* contact states hand instances during training.

**Evaluation Metric.** We measure the joint hand detection and contact recognition performance using VOC average precision metric. We consider a detected hand instance to be a true positive if: (1) the Intersection over Union (IoU) between the axis parallel bounding box of the detected hand and a ground truth bounding box is larger than 0.5; and (2) the predicted contact state matches the ground truth. More precisely, for each contact state, we only consider hand boxes annotated *Yes* for that contact state to be ground truth boxes. We then measure the joint hand detection and contact state recognition AP by multiplying the hand detection score with the predicted contact score. We do not measure the performance for detections that have overlap with *Unsure* contact state hand instances.

**Baselines.** Given the hand's location, one might think of learning a classifier on such hand crops to obtain its contact state. To see how such a method performs, we train ResNet-101 based classifiers on hand crops from the training set of the ContactHands dataset. We consider two types of hand crops, one corresponding to the hand's axis-parallel bounding box, and another corresponding to the quadrilateral bounding box. To construct a rectangular image from a quadrilateral box, we first obtain a rotated rectangular bounding box and then build an axis-parallel image crop. We further consider two variants for each type of hand crop, the exact bounding box and the extended bounding box to provide surrounding context information. To obtain this extended bounding box, we increase each side of the hand crop's length by 50% so that the expanded bounding box has an area 2.25 times the original bounding box area. Altogether, there are four variants, and we train four different

| Contact State | Axis-Parallel | | Quadrilateral | | Pose Heuristic |
| --- | --- | --- | --- | --- | --- |
| | Exact | Extended | Exact | Extended | |
| No-Contact | 24.49 % | 45.82 % | 40.90 % | 29.27 % | 35.50 % |
| Self-Contact | 24.76 % | 38.57 % | 34.63 % | 30.16 % | 38.29 % |
| Other-Person-Contact | 3.49 % | 3.99 % | 4.11 % | 3.92 % | 4.48 % |
| Object-Contact | 51.63 % | 65.88 % | 62.19 % | 58.53 % | 61.30 % |
| mAP | 26.10 % | 38.56 % | 35.46 % | 30.47 % | 34.89 % |

Table 1: **Hand contact recognition APs** of ResNet-101 classifiers and a method based on human pose estimation. The performance is evaluated on the test set of the ContactHands dataset.

| Method | M-RCNN | Proposed | M-RCNN | Proposed | Proposed | Proposed | Proposed | |
| --- | --- | --- | --- | --- | --- | --- | --- | --- |
| Train data | 100DOH | 100DOH | C-Hands | C-Hands | 100DOH | C-Hands | 100DOH + C-Hands | |
| Test data | 100DOH | 100DOH | C-Hands | C-Hands | C-Hands | 100DOH | 100DOH | C-Hands |
| No-Contact | 67.30 % | 68.23 % | 60.52 % | 62.48 % | 44.45% | 47.13 % | 70.16 % | 63.90 % |
| Self-Contact | 54.94 % | 58.52 % | 51.62 % | 54.31 % | 32.03 % | 38.85 % | 58.66 % | 59.30 % |
| Other-Person | 6.56 % | 12.94 % | 33.79 % | 39.51 % | 7.32 % | 6.77 % | 21.15 % | 42.01 % |
| Object-Contact | 90.34 % | 92.70 % | 67.43 % | 73.34% | 49.68 % | 74.27 % | 88.21% | 70.49 % |
| mAP | 54.78 % | 58.10 % | 53.31 % | 57.41 % | 33.37 % | 41.76 % | 59.54 % | 58.93 % |

Table 2: **Joint hand detection and contact recognition APs** using different methods and datasets. M-RCNN denotes Mask-RCNN. 100DOH denotes video frames dataset [27] and C-Hands denotes our dataset ContactHands.

ResNet-101 classifiers and evaluate contact recognition performance on the test set of ContactHands. We summarize the results in Table 1. These results show that learning a classifier directly on hand crops is not adequate for recognizing their contact states.

Given the success of 2D human pose estimation methods, we want to know if we can use the relationship between a hand and joint locations of humans to reason about the hand's contact state. For this purpose, we build a feature vector $\mathbf{h} \in \mathbb{R}^{52}$ for each hand instance using the following heuristic. We use [4] to detect keypoints corresponding to 25 human joints. Additionally, we use an object detector to detect all possible objects in the scene. Then for each hand instance, we build three types of features. First, we construct a vector $\mathbf{h_s} \in \mathbb{R}^{24}$ of distances from the wrist joint to the other 24 joints of the same person. Second, we obtain a vector $\mathbf{h_p} \in \mathbb{R}^{25}$ of average distances from the wrist joint to 25 joint locations of other people in the scene. Here, the average is with respect to other people. Finally, we obtain a vector $\mathbf{h_o} \in \mathbb{R}^3$ encoding the relationship between the hand and the detected objects. Precisely, the first component of $\mathbf{h_o}$ is the mean distance of the hand from the detected objects. The second and third components of $\mathbf{h_o}$ are the mean overlap, and the mean IoU of the hand with the detected objects. We obtain the final feature vector $\mathbf{h} \in \mathbb{R}^{52}$ for the hand by concatenating $\mathbf{h_s}$, $\mathbf{h_p}$ and $\mathbf{h_o}$. We use such hand feature vectors $\mathbf{h}$ to train a classifier on the training set of ContactHands. The last column of Tab 1 summarizes the performance of the classifier on the test set of ContactHands. The results show that human pose heuristic methods are insufficient to estimate hands' contact states in unconstrained conditions.

**Main Results.** We now present the proposed method's results and compare it to Faster-RCNN and Mask-RCNN to detect and recognize hand contact states. For this purpose, we use the training and test splits from the ContactHands dataset and the 100DOH [27] dataset. The 100DOH is a video-frame dataset and has 79,920 training images and 9,983 test images. We conduct experiments by training the proposed architecture and compare it to a modified Mask-RCNN that can detect hands and recognize contact states. We summarize the results of these experiments in Table 2.

The proposed method has better performance than the Mask-RCNN since it considers surrounding objects into account when making a contact decision. We also experimented by using Faster-RCNN instead of Mask-RCNN, and it performs similarly to Mask-RCNN. Specifically, we found that Faster-RCN has 54.04 % mAP when trained and tested on the 100DOH dataset and 53.23 % mAP when trained and tested on the ContactHands dataset.

The fifth and sixth columns show the cross dataset evaluation performance. We can see that a model trained on the ContactHands dataset has better cross dataset generalization performance than the

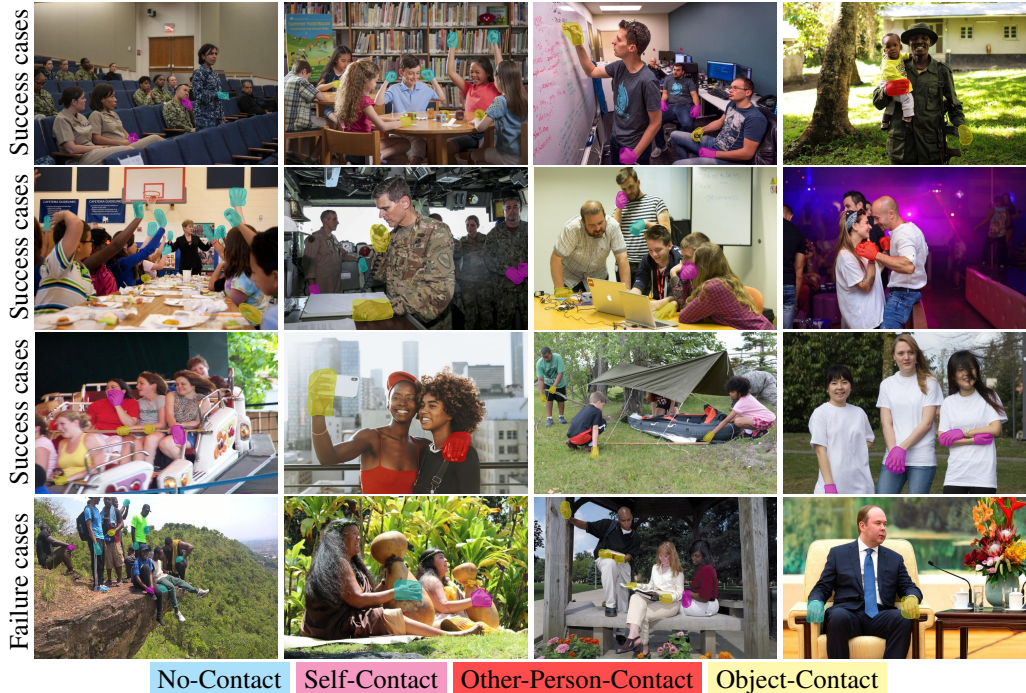

Success cases
Success cases
Success cases
Failure cases

No-Contact  Self-Contact  Other-Person-Contact  Object-Contact

Figure 5: **Qualitative results and failure cases.** The first three rows show some good qualitative results, and the last row shows some failure cases from our method. We visualize detected hand instances by their predicted contact state color. We add additional contact state labels if a hand is in more than one contact state.

100DOH dataset model. These results show the benefit of our data. The last two columns show that a model trained on a combination of the 100DOH dataset and the Contact-Hands data performs better than models trained on individual datasets separately.

**Ablation Studies.** We conduct experiments to study the effect of different components of the Contact-Estimation Branch. Specifically, we train the proposed network on the training set of ContactHands by removing the cross-feature affinity-based attention module, the spatial attention module, and both. We evaluate these methods on the test set of ContactHands, and they achieve 56.08%, 55.91%, and 55.12% mAP on the joint task of detecting hands and recognizing their contact. Comparing these results with the full architecture that has 57.41% mAP shows that both the attention methods are useful for estimating hand's contact state.

**Qualitative Results and Failure Cases.** Figure 5 shows some qualitative results from the proposed model, trained on the ContactHands dataset. The first three rows show good results, and the last row shows failure cases. The failure cases are mainly from two sources, false hand detections and bad contact state predictions. First, sometimes other skin areas are being mistaken for hands, and thus hand detections are not perfect. Second, even if a hand is detected correctly, it's predicted contact state can be incorrect; when a hand is surrounded or occluded by other objects, the lack of depth information can make the contact decision challenging.

# 6   Conclusions

We investigated a new problem of hand contact recognition. We introduced a novel Contact-Estimation neural network module that can be trained end-to-end with any two-stage object detector to detect hands and recognize their physical contact states simultaneously. We also collected a challenging large-scale dataset of unconstrained images annotated with hand locations and their contact states. Hand contact recognition is a less-explored problem with important applications. It is also a challenging problem, especially in unconstrained environments, and there is massive room for improvement. We hope our work will further spark the community's interest in addressing this important problem.

# 7   Broader Impact

We can broadly classify hand contact estimation methods into two types, contact recognition, and contact detection. While the contact recognition methods categorize hand instances into pre-defined states, contact detection methods aim to detect contact objects and contact areas. In our work, we investigated hand contact recognition, particularly in unconstrained images. We believe that our work can help the community accelerate the research in this area and shed light on contact detection. While our dataset is quality controlled and beneficial to the community, biases can be present in the sources from which we collected the dataset. For example, the dataset might not be representative of all demographics. As such, applications that use our dataset can inherit such biases, and the community should be aware of this.

**Acknowledgement.** This research is partially supported by US National Science Foundation Award IIS-1763981, the SUNY2020 Infrastructure Transportation Security Center, and VinAI Research.

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
