[Reviews · NeurIPS 2020]

Review 1

Summary and Contributions: The paper considered the problem of detecting hands in RGB images and classifying their contact state (no contact, self-contact, other-person-contact, object-contact). As key contributions, the authors list a new large-scale dataset and two attention mechanisms (cross-feature affinity attention module + spatial attention mechanism) that allow training from weak annotations only (in this case no hand or body part annotations). These mechanisms are inspired by related work. In order to train and test their network, the authors created a 21,637 image dataset with hand segmentation and contact condition. They achieve 57.41% mean average precision on a joint hand detection and contact state metric. The authors have very extensive experiments that dissect how much improvement each of their contributions bring. This includes a comparison to existing work.

Strengths: The problem of detecting physical contact states seems very relevant with applications ranging from e.g. harassment detection, activity recognition and contamination prevention. The empirical evaluation of the approach is sound with a range of ablation studies, and comparison to prior work as well as simple heuristics to motivate the author’s model design choices. The proposed method always performs somewhat better. The work makes clear that hand segmentation passed through a classifier alone is not sufficient to do contact classification. They also show that detecting a human body joint model and performing overlap with objects in the scene also does not work well. The dataset seems very useful to the community and seems to be thoroughly quality controlled. I would like to thank the authors for creating this dataset which takes much work.

Weaknesses: First of all, it is confusing that the authors claim to compare against [28] while in that paper, the model is based on FasterRCNN and bounding boxes instead of MaskRCNN with an additional head for contact classification as claimed here. This puts the evaluation results to this work into question. -> The authors have clarified this in their rebuttal. Assuming that the authors have correctly compared against [28], it seems that 4% improvement on average over all the contact state is relatively low. However, the method consistently outperforms the baseline on each of the categories even when it is trained on the new dataset. However not by a large margin. -> I may be miss-caibrated on what a significant contribution constitutes. There is a 10% drop in performance of the prior model when evaluated on the new dataset but trained on the old one. This may be due to overfitting to the other dataset. What is missing is an evaluation of the authors’ method trained on the previous dataset [28]. This would make it clear why the collection of a new dataset is necessary. [28] make the choice to use video frames to capture natural hand movement. It is unknown how the current work does in such a context. The authors should include both of the following * The new algorithm trained on the video frames from [28]. This is necessary as a comparison to column 1 of table 2. * A comparison of the video results versus still image results with this algorithm. After Rebuttal: I appreciate that the authors have conduced all the experiments. It seems though that their approach trained on data from [28] performs better than when trained on their data (significantly except for other-person). However, the authors do show that transfer of the model is better. These results puts the dataset contribution into question. It is unclear how to evaluate the dataset contribution given the mixed results. The ablation study also shows that removing any single part of their network had little change on their results. It would be good to understand better why that is. -> the following two points were also addressed in the author's rebuttal and are no longer a weakness There is substantial work in detecting hand joint locations, which seem like a more appropriate comparison than human body joints. It is unclear on why the authors made the choice to use a full body model rather than hand joints. [Simon, Tomas, et al. "Hand keypoint detection in single images using multiview bootstrapping." Proceedings of the IEEE conference on Computer Vision and Pattern Recognition. 2017.] [https://ai.googleblog.com/2019/08/on-device-real-time-hand-tracking-with.html] [Armagan, Anil, et al. "Measuring Generalisation to Unseen Viewpoints, Articulations, Shapes and Objects for 3D Hand Pose Estimation under Hand-Object Interaction." arXiv preprint arXiv:2003.13764 (2020).] It is additionally unclear why the joint location baselines would use mean overlap and IoU of the hand with objects. The maximum could be more successful, since the object with the maximal overlap seems most relevant.

Correctness: The paper seems correct except that they are incorrect about what they claim the baseline model is [28] (FasterRCNN versus MaskRCNN). I am not an expert in attention and therefore cannot judge the novelty and technical correctness of these claimed technical contributions.

Clarity: The paper is clearly written and easy to follow, but it is suggested to have the paper further proofread for english style (example place for improvement line 135: “...... both the hand and the object, and also …..” ) It is not clear what the colors used in Figure 2 are trying to convey. The statistics of the dataset could be better visualized. For example, a bar plot can better represent the relative number of hand instances for the 4 different contact states Typo in line 61: “……. a cross-feature affinity attention module *** that*** can combine hand…..” Text overlay in figure 4 is very hard to read. Try using a different color. Figure 1 has a lot of unused space. The context estimation branch box is too small. Figures 2, 3, 4 are not referenced in the text. Would be easier to read if phi, psi, and theta appeared in their respective subsections, instead of just in 3.4. It might actually be more clear if 3.4 came before 3.3 and 3.2.

Relation to Prior Work: The related work is extensive but stays somewhat superficial when comparing. It wasn’t clear exactly which aspects of the architecture and framework were completely novel and which ones were simply novel applications of prior work. Perhaps it would help if the Related Work section dedicated more description to important relevant works. It is not fair to say that the algorithm of Shan et al “performed poorly” in your Related Work section, as its performance was quite near yours when trained on the same dataset, as shown in Table 2. Please assuage that language. [Contextual Attention for Hand Detection in the Wild, Narasimhaswamy et al, 2019] seems to have a particular relevant background and similar method for using maskRCNN for hand segmentation, but isn't called out specifically. The algorithmic and dataset similarity to [Understanding Human Hands in Contact at Internet Scale, Shan, 2020] could have used elaboration as well as already mentioned above. Including how the algorithm fares when trained on this dataset are missing.

Reproducibility: Yes

Additional Feedback:


Review 2

Summary and Contributions: This papers addresses the task of inferring the physical contact state of hands from a single image. Specifically, the task is to localize each hand instance with a bounding box and mask, and classify the hand into four contact states: "no-contact", "self-contact", "other-person-contact", and "object-contact". To solve this task, this paper proposes a ConvNet with two special attention modules ("cross-feature affinity attention module" and "spatial attention module"). The paper also introduces a new dataset of in-the-wild images with contact state annotations. Finally, the proposed method is evaluated on the proposed dataset and achieves a higher classification mAP over several baselines.

Strengths: [+] The addressed task is definitely interesting and important. It is also very relevant in the context of the current pandemic situation. [+] Compared to hand detection and pose estimation, contact state estimation seems to be less explored in the literature. This problem certainly deserves more attention and just on that note I think this papers deserves a promotion due to its topic. [+] The paper also contributes a new dataset and that might be valuable to the research community. It's also worth noting that [28] has also introduced a dataset of similar kind but I assume they had not released it to public at the time this paper was submitted?

Weaknesses: [-] The first weakness of this paper is on the technical contribution. I did not see any surprisingly new ideas in the methodology. The overall architecture of the network largely follow the typical paradigm of object detection and classification (e.g. proposal generation, RoI pooling, N-way classification, etc.). The loss functions (e.g. classification, bounding box regression, etc.) are also very standard. [-] The main feature of the approach seems to be the two novel attention modules. However, the mechanism of these attention modules are also fairly standard (e.g. weight prediction, element-wise multiplication, etc.). The main difference could be on how these two modules are connected (i.e. "Contact-Estimation" branch in Fig. 1). However the design seems a bit random: I did not find it very intuitive and I don't see much explanation on why these modules are put together this way. In other words, there are many different ways to fill in this "Contact-Estimation" block and why this one? *** This was addressed in the author's rebuttal (L21). [-] While the dataset may be valuable due to its novelty, the size of the dataset is rather limited given today's standard: a total of 18K training images and only 1.6K test images. *** This was addressed in the author's rebuttal (L34). [-] On the detail of the data annotation process: Why localizing hand instances with quadrilateral boxes (L219)? Why can't it be a polygon with any possible number of vertices, so that you don't limit the accuracy of these annotations? *** The authors attempted to address this in the rebuttal (L27), but the response was not satisfying. For "small and blurry hands", the annotator still has the choice of using a polygon with few vertices. My point was that using a free form polygon and not constraining to quadrilaterals offers better flexibility. [-] There is also some issues on the experimental comparison. From L290, it seems that the first column of Tab. 2 represents the results of [28]. Why not explicitly add the citation label [28] to that column but rather than calling it "Mask-RCNN"? If it does not represent [28], what is the difference then? *** This was addressed in the author's rebuttal (L29) and R1's comments. [-] Following the last point, it is also unfair to compare with [28] by only running inference with their trained model (L290: "evaluating Shan et al. [28]'s model"). Why not try to re-train their model with the new data and compare with that? *** This was addressed in the author's rebuttal (L29).

Correctness: I have issues with the following claim (L44-47): "there is no annotation indicating which object or parts of an object a hand is in contact with, and one technical contribution of our paper is a method to train the Contact-Estimation module using data that only has weak annotation". By using the word "weak annotation", the paper implicitly assumes that "predicting object or parts of an object that is in concat" is part of the task. However this is not the case, at least I have not seen that evaluation (predicting the contact object) anywhere in the paper. *** This was addressed in the author's rebuttal (L31).

Clarity: Yes, the paper is clear and easy to read.

Relation to Prior Work: Yes, this is discussed in the related work section.

Reproducibility: Yes

Additional Feedback: * Suggestions: - L51-52: "we obtain ... using generic object detector pretrained on the COCO dataset" -> Does this mean the model can only handle contact with COCO objects? If so, the paper should explicitly point this out. - Eq. 1: "Z" is not used again anywhere later in the paper. It is renamed to \phi in L171? - L168: Abuse of notation "i" since "i" has already been use as index of self attention maps in Sec. 3.3. - Looking at some of the sample images (Fig. 3), I feel that the task is either trivial (e.g. holding an object, in which case the contact state is always true) or super difficult without depth (e.g. determine whether A is touching B when A is in front of B from the camera view point, such as in the rightmost figure of Fig. 3). Essentially we are solely relying on semantic information here. This observation is also supported by the highly imbalanced mAPs in Tab. 1. I think this is an interesting insight and probably worth some discussions in the paper. - Fig. 4 is not mentioned anywhere in the text. - Fig. 4: "shorthand text overlaid on the hand mask" -> These texts are illegible on printouts. * Questions: - Tab. 1: For "Axis-Parallel", "Extended" performs better than "Exact", while this is the opposite case in "Quadrilateral". Any explanation? *** This was addressed in the author's rebuttal (L38). - For the AP metric (L251), is the IoU computed from the quadrilateral box or the axis-aligned box? *** This was addressed in the author's rebuttal (L42). ##### Post-rebuttal As noted in my initial review, one main concern is on the comparison with [28], both on the dataset front and the methodology front (R1 also had similar concerns). I found Tab. 1 from the rebuttal particular helpful in clarifying the concerns. On the algorithm front, it shows the comparison to [28] under same training/test settings. On the dataset front, it provides cross-dataset results. I'd encourage the authors to include these results to the paper. Overall, while this is not a super strong submission, I do find some merits in this paper (a new dataset and a method with sufficient edge over prior approaches). With that I'm leaning towards accept in my final opinion.


Review 3

Summary and Contributions: This paper investigates a new problem of detection hands and estimate their physical contact state with surrounding environment. Along with the new problem, the paper also proposed a new dataset, 22k images, with good annotation. Each hand's state can be one of the four states, including No-contact, Self-contact, other-person-contact, and object-contact. With the new dataset, the paper also proposed a new model, based on Mask-RCNN, to jointly localize hands and predict their physical state.

Strengths: This paper investigates a new problem, proposes a new dataset, and a new method based on Mask-RCNN. Compared with the baseline (Mask-RCNN) the proposed method improve it from 53% to 57%. The proposed Contact-Estimation branch includes two types of attention mechnisms. One is based on the affinity between the hand and a larger region. The other on is used to adaptively select discriminative features from a plausible region of contact.

Weaknesses: Is the dataset balanced? Other-Person-Contact seems to be more difficult? As shown in the experiments, Table 1 and 2. In table 1, the accuracy of object-contact is 20 times higher than other-person-contact. In table 2, the accuracy of object-contact is almost 2 times higher than other-person-contact. Is it because of not enough training data? or this contact state too difficult? It's hard to understand the second sentence of figure 1's title? Does it mean the feature of hand, the feature fo the object and feature of the union box? Broader impact is well written, what would be the negtive impact? can this technology lead to privacy issues?

Correctness: looks correct

Clarity: well written

Relation to Prior Work: yes

Reproducibility: Yes

Additional Feedback: *****after rebuttal I have carefully read the authors rebuttal and other reviewers comments, and still keep the original rating.

[Author Response · NeurIPS 2020]

Table 1: Comparing FasterRCNN, MaskRCNN and proposed method, and cross-dataset experiments.

| Method | FasterRCNN [28] | MaskRCNN | Proposed | FasterRCNN [28] | MaskRCNN | Proposed | Proposed | Proposed |
|---|---|---|---|---|---|---|---|---|
| Train data | [28] | [28] | [28] | ours | ours | ours | [28] | ours |
| Test data | [28] | [28] | [28] | ours | ours | ours | ours | [28] |
| No-Contact | 66.55 % | 67.30 % | 68.23 % | 59.22% | 60.52 % | 62.48 % | 44.45% | 47.13 % |
| Self-Contact | 53.45 % | 54.94 % | 58.52 % | 50.96 % | 51.62 % | 54.31 % | 32.03 % | 38.85 % |
| Other-Person | 6.46 % | 6.56 % | 12.94 % | 32.00 % | 33.79 % | 39.51 % | 7.32 % | 6.77 % |
| Object-Contact | 89.70 % | 90.34 % | 92.70 % | 70.75 % | 67.43 % | 73.34% | 49.68 % | 74.27 % |
| mAP | 54.04 % | 54.78 % | 58.10 % | 53.23 % | 53.31 % | 57.41 % | 33.37 % | 41.76 % |

**Reviewer 1:** **Q1:** Are the evaluation results valid since [28] uses FasterRCNN instead of MaskRCNN? The paper
should include an evaluation of the proposed method trained on the previous dataset [28]. **A:** In the context of
physical contact estimation, there are no conceptual nor empirical differences between MaskRCNN and FasterRCNN.
Conceptually, MaskRCNN is FasterRCNN with an additional mask prediction branch. Empirically, they perform
similarly as shown in Tab 1. All results are evaluated using bounding boxes. The data from [28] was not available at the
time of submission. The experiments requested by the reviewer are now shown in Tab 1. The cross-dataset results from
the last two columns show that the model trained on our data has better cross-dataset generalization (by 8% in mAP)
when compared to the model trained on the previous dataset [28]. This also shows the benefit of our data.

**Q2:** 4% improvement on average over all the contact states is relatively low. **A:** We respect your opinion. But an
average improvement of 4% is significant, and a higher experimental standard would have not been satisfied by many
previously published NeurIPS papers.
**Q3:** In the ablation study, removing any single part of their network had little change on their results. **A:** Removing
any single component reduces mAP roughly by at least 1.5%, while removing both components reduces mAP by 2.5%.

**Q4:** Hand joint locations seem more appropriate comparison than human body joints. **A:** Following the suggestion, we
used OpenPose [5] for hand keypoints, but it failed to detect hands in many unconstrained images, as also reported in
[22]. Empirically, we found that the detection AP is only 39.36%, compared to 83.72% of our method. This level of
noisy detection results cannot be used for contact state estimation.

**Q5:** For the joint location baselines, why use Mean overlap of the hand with objects, instead of Maximum? **A:** While
Maximum seems to be more intuitive, it does not perform better in practice, yielding an mAP of 33.73%. We originally
used Mean because it was thought to be more robust than Max for noisy inputs (i.e., noisy detection results).

**Reviewer 3:** **Q1:** Why are two attention modules connected this way? **A:** First, the region between hand and object
can have plausible regions of contact and we want to predict contact scores directly by spatially attending such regions
using the spatial attention module. Second, the appearance of the hand and its affinity between surrounding objects
provides strong cues in determining its contact state. We encode these information using cross-feature affinity attention
module and predict another set of scores. The Contact Estimation branch is designed to combine two sets of scores
from two attention models.
**Q2:** Why are hand instances annotated with quadrilateral boxes instead of any number of vertices? **A:** Due to many
small and blurry hands in our data, it would be ambiguous and prohibitively laborious to use free form polygons.

**Q3:** Does the the $1^{st}$ column of Tab 2 represents the results of [28]? Why not adding the ref [28] to that column instead
of calling it Mask-RCNN? Retrain [28] with new data **A:** Yes, this will be fixed. See Tab 1 for requested experiments.

**Q4:** By using "weak annotation", the paper implicitly assumes that predicting the object is part of the task. **A:** We
understand your concern, and we will clarify it in our revised paper. We adopted the term "weak annotation" from the
field of multiple instance learning, where detection is not necessarily the main task.
**Q5:** The dataset of 22K images appears to be small. **A:** We respect your opinion. But 22K images is not small.
Besides, our dataset is challenging and diverse, and has many images where it is not trivial to estimate contact states.
For instance, the results from the last two columns of Tab 1 shows that our dataset performs much better when compared
to the previous larger dataset in the task of cross dataset evaluation.
**Q6:** For Axis-Parallel, Extended performs better than Exact, but is the opposite case in Quadrilateral. **A:** When using
Quadrilateral, to crop the polygon into a rectangular image, we first construct a rotated rectangle. Because of this, some
surrounding context region is already present. Extending this even more can add a lot of irrelevant regions and leads to
a reduction in performance.
**Q7:** For the AP metric, is the IoU computed from the quadrilateral box or the axis-aligned box? **A:** We used
axis-aligned box, following the standard evaluation protocol for hand detection [1, 22, 28].

**Reviewer 4:** **Q1:** Is the dataset balanced? Other-Person-Contact seems to be more difficult? **A:** We aimed for a dataset
of representative images of the real world, so the classes are not balanced. Compared to other classes, the number of
hands with Other-Person-Contact labels is much smaller, as reported in lines 236–241.
**Q2:** For the second sentence of Fig 1, does it mean the feature of hand, the feature of the object and feature of union
box? **A:** It means the feature of hand, and the feature of the union box. We will reword this, thanks.

[Meta-Review · NeurIPS 2020]

The initial scores for this paper were: 4: An okay submission, but not good enough; a reject. 6: Marginally above the acceptance threshold. 6: Marginally above the acceptance threshold. This is a borderline case. The positive points are: - The work addresses an important and less explored problem of detecting physical contact states. - The work provides sound empirical evaluation with a range of ablation studies, comparison to prior work and well-chosen baselines. - New dataset with good quality control that will be useful for the community. The negative points: - Relatively small improvements over the state-of-the-art (4%), though the improvements are consistent. - Missing experiments. - Some clarity issues. - Better description of relation to prior work is needed. - Limited technical novelty. - Relatively small size of the dataset (18k training images, 1.6k test images). The authors provide a rebuttal, which addresses many of the weak points. In the post-rebuttal discussion, the negative reviewer R1 (4) upgrades their score from 4 to 6. R3 confirms their positive rating. The final scores are 6, 6, 6 with all reviewers leaning towards acceptance. The AC is convinced by the positive arguments of the reviewers and recommends Accept. The authors are strongly encouraged to take into account all reviewers' feedback when preparing the final version.